# Distributed High Temperature Monitoring of SMF under Electrical Arc Discharges Based on OFDR

**DOI:** 10.3390/s20226407

**Published:** 2020-11-10

**Authors:** Chen Chen, Song Gao, Liang Chen, Xiaoyi Bao

**Affiliations:** Department of Physics, University of Ottawa, 25 Templeton Street, Ottawa, ON K1N 6N5, Canada; cchen340@uottawa.ca (C.C.); sgao044@uottawa.ca (S.G.); Liang.Chen@uottawa.ca (L.C.)

**Keywords:** optical fiber sensors, optical frequency-domain reflectometry, Rayleigh backscattering, distributed temperature sensing

## Abstract

The distributed high temperature measurement of an optical fiber subjected to electric arc discharges based on optical frequency-domain reflectometry is experimentally demonstrated. The distributed temperature profile is attained in an open glow regime of a few milliamps with maximum detectable temperature up to 2100 ± 20 °C. The discharge arc-induced softened length of the fiber and mechanical stress are measured and statistically analyzed in terms of the correlation of the Rayleigh spectra. The large wavelength scanning range of OFDR enables much higher accuracy for the delay time measurement with a minimum measured delay of 40 fs. The delay shift over the entire heating range for a single discharge duration is statistically calculated by using a temporal correlation method. The reliability of the thermal sensitivity coefficient as 10 pm/°C for telecom single mode fiber (SMF, @1550 nm) is quantitatively analyzed and evaluated by the correlation coefficient. Lastly, a spectral mapping method is employed in spectrum monitoring for discharge dynamic impact on the optical path length (OPL) and local Rayleigh scatter.

## 1. Introduction

Optical frequency-domain reflectometry (OFDR) provides both the high sensitivity and high spatial resolution necessary to acquire the localized Rayleigh scatter pattern. It has been widely applied to distributed temperature and strain measurement [1,2]. The basic working principle is based on the assumption that the amplitude and phase pattern of local Rayleigh spectra have a static random property over the fiber length, which can be modelled as a long, weak fiber Bragg grating (FBG) with a random period [3]. Several research works have been reported regarding high splicing temperature measurement in electric arc discharges. I. Hatakeyama et al. used a microradiation thermometer to obtain the fiber core temperature up to 2000 °C while investigating the fusion splices for single mode fiber [4]. By relating the internal stress relaxation to the temperature and annealing time of optical fibers, Y. Mohanna estimated the electric arc temperature at around 1450 °C, with a Gaussian distribution [5]. G. Rego et al. used blackbody radiation to determine the temperature of a small piece of fiber exposed to an electric arc discharge at 1450 °C [6]. Afterwards, they used an electrically insulated Type S thermocouple to measure the temperature of an optical fiber while being heated through electric arc discharges at 1400 ± 50 °C [7]. The difficulty in estimating extremely high temperatures within a narrow spatial region, such as the heating environment created by the splicing arc of sub-mm makes the OFDR sensor to be the best candidate for distributed high temperature measurement in an open thermal equilibrium system. The temperature in the fiber during the electric arc discharge is an important parameter, as it is related to splicing loss, fabrication of gratings with different reflection, and hence, optimization of the fusion splicer. It would be interesting to have an insight of the distributed thermal radiation and convection generated by the electric discharge on the fiber by taking advantage of the high spatial/temporal resolution of OFDR.

In this study, we present a feasibility study inside a fusion splicer using telecom fiber based on OFDR with a high spatial resolution of 8μm over a less than 10-mm length of the fiber by measuring the thermal-induced wavelength shift in the reflected spectrum of the Rayleigh backward scattering (RBS) [3]. The results presented herein demonstrate the capability of the distributed ultrahigh temperature measurement based on OFDR, with the possibility of maximum detectable temperature for optical fiber up to 2100 °C created by an electrical arc. The 10 pm/°C as temperature sensitivity coefficient is maintained at an ultrahigh temperature gradient with a Gaussian-like profile. Based on [8] the viscosity, the vitreous silica determines the softening and annealing temperature of the material. By referring to the variation of viscosity as a function of temperature, we can evaluate transient temperature near the phase changing condition. The temperature is quantitatively evaluated by zero-mean normalized cross-correlation (ZNCC), which compares the similarity of Rayleigh spectra of the same fiber location under heating and cooling conditions. In addition, internal stress induced by the high temperature gradient of discharge is statistically analyzed at different arc currents. The thermally induced group delay shift of the Rayleigh pattern is observed by correlating reference and measurement traces in time domain based on OFDR. We attribute this to the thermal energy transferred from ARC current flow to the material of vitreous silica via ultraviolet absorption and thermal convection, which initially leads to the transition of electrons from ground state to excited state and the thermal motion of the lattice, and hence changed the optical path length (OPL). A multiple delay shift induced by coexisting phase modes was also determined by employing this method, which provides a new assessment tool on the relative delay of the scattering response. Lastly, dynamic monitoring of the discharge impact on the OPL and intensity change in the Rayleigh spectra with time over 1 s is presented using OFDR. Different from OTDR with time sampling due to the repetition rate which missed some time dependent information, wavelength scanning in OFDR is superior for dynamic ARC measurement, which enabled continuous measurement of delay time induced by the refractive index change of the fiber melting in the ARC process. Combined with a spectral mapping method, it provides continuous tracking for intensity change over spatial and spectral dimension. The OTDR technique recovers phases over small wavelength change while the refractive index change over small wavelength range can be neglected, often a few pm range, and hence accuracy of the relative time delay change is limited to a fraction of the pulse width (a few ns). However, the refractive index change over wavelength range of 100 nm with OFDR enables much higher accuracy for the delay time measurement with minimum measured delay shift of about 40 fs, when the rest of the fiber has no change in refractive index, one can attribute the measured delay to the arc process in the sub-mm region, which increases time resolution for dynamic measurement.

## 2. Principle and Experimental Setup

The experiment setup designed for distributed high temperature measurement in a commercial fusion splicer (Ericsson, Sweden) is shown in Figure 1. Initially, a tunable laser source (TLS, New-Focus-Venturi-TLB-6600, Newport Corporation, Irvine, CA, USA) was coupled into the fiber and split the light into two parallel branches of Mach-Zehnder interferometer, one of which functioned as the auxiliary interferometer (AUI), circled by a dash square. The output of AUI played as an external clock for data acquisition, a known frequency-sampling method and widely adopted for its accuracy and convenience in coping with the nonlinearity of the laser tuning [9]. Another branch named as main interferometer (MI), was used for the interrogation of the fiber under test by splitting the injected light into reference and probe arms. In this setup, light propagating in the probe arm was further guided by a circulator and returned in the form of the backward Rayleigh scattering, delayed and encoded with dynamic discharge action. Reflected probe fields then recombined with the reference fields and created detectable beat patterns as interference signal, eventually recorded by photodetectors and digitalized for post data processing. A polarization beam splitter (PBS) and a polarization controller (PC) were used to balance the reference light evenly allocated into two orthogonal polarization states, labeled as S state and P state. This polarization diversity technique is used in OFDR to mitigate signal fading due to polarization misalignment of the interfering measurement and local-oscillator fields [10]. Considering the wavelength tuned linearly by the laser, light propagating in the reference arm arrives at the photodetector prior to that in the probe arm, which leads to a delay between the two arms and hence the generation of the beat frequency corresponding to the delay. Essentially, for a single wavelength, the delay of the light propagating in a single mode fiber (SMF) depends on the ratio of the speed of light in vacuum to the product of local refractive index n and its physical length L. The dispersion in a short length of SMF can be negligible over a small range of wavelength, the local refractive index of SMF can be replaced by a global constant to locate the position under high temperature. The spatial resolution in Rayleigh scattering measurement is determined by the spectral bandwidth of the tuning range according to:$$\Delta z=\frac{\lambda_{s}\lambda_{e}}{2n_{g}\Delta \lambda }$$
where $n_{g}$ is the group index of the fiber under test, holding with an assumed constant value of 1.47. $\lambda_{i}$ and $\lambda_{f}$ are the initial and final wavelengths, Δλ is the total scanned wavelength range in the measurement. In this setup, the laser scanned from 1520 nm to 1620 nm, which corresponds to a spatial resolution Δz approximate to 8 μm. However, the thermal-induced refractive index change and fiber expansion makes the former approximation of refractive index no longer valid considering the locally nonuniform change in OPL under arc. The 1D Maxwell equation in an inhomogeneous and time varied medium due to the relative permittivity changing with position and time along the propagation direction can be written as:$$\frac{\partial^{2}E_{z}}{\partial z^{2}}+\frac{\partial }{\partial z}\left(\frac{1}{\varepsilon }E_{z}\frac{\partial \varepsilon }{\partial z}\right)=\frac{1}{c^{2}}\left[\varepsilon \frac{\partial^{2}E_{z}}{\partial t^{2}}+2\frac{\partial \varepsilon }{\partial t}\frac{\partial E_{z}}{\partial t}+E_{z}\frac{\partial^{2}\varepsilon }{\partial t^{2}}\right]$$

If we further consider a spatial nonuniformity and assume static waveguide, ∂ε/∂t=0, and ignore the time and wavelength dependence on the local refractive index change, which is valid for an open thermal equilibrium state over a small range, we can simplify the refractive index as a function of position, and thermally induced optical path length $\tilde{\Gamma }\left(z\right)$ on Rayleigh scattering response at position *z* as:$$\tilde{\Gamma }\left(z\right)=\int_{0}^{z}\sqrt{\bar{\varepsilon }+\delta \varepsilon \left(\tilde{z}\right)}d\tilde{z}$$
where $\bar{\varepsilon }$ and δε are the average relative permittivity and its spatial fluctuation, respectively. Basically, the delay information is the result of a spatially resolved interference pattern via Fourier transformation. Note a delay shift of Rayleigh pattern occurs under the arc condition due to the increased OPL that was measured in reference to room temperature.

## 3. Experimental Results and Data Analysis

### 3.1. Distributed Temperature Profile Varied with Currents

In the splicer used, the electrode gap length and fusion time were fixed and the arc current varied, which determined the overall glow heating temperature. A stripped fiber held in embossed grooves was aligned and clamped on the fusion stage and we ensured that it was immobile during the multiple discharges without any external pulling or pushing forces applied. Changes caused by an external stimulus (such as strain or temperature) in turn gave rise to shifts in the locally reflected spectrum of the Rayleigh scattering. As the spectrum shift was proportional to the stimulus applied on the fiber, these local spectral shifts could then be calibrated and applied to a distributed strain or temperature measurement. To begin with, the measurement of the Rayleigh scattering signature was accomplished at almost the last second of the discharge by artificially controlling the commencement of data collection, then followed by measuring the signature at room temperature, which was taken as a reference. By correlating the Rayleigh spectra measured at two different thermal conditions, the related wavelength shift was then determined and hence relative temperature variations.

In our setup, the tuning rate of the light source was set to 100 nm/s, hence the auxiliary interferometer equipped with a 24.3 m length of delay fiber generated a sampling rate of 1.55 MHz for Digital Acquisition (DAQ). Every single trace, either for S or P polarization state, took up to 1.03 s, being detected by photodetector (Thorlabs-PDB130C) and eventually generated into a 1.6 mega-sample of raw data by digital acquisition card (NI-PCI6115). The splicer with electric current between 4.0 mA to 6.0 mA was applied to the fiber. It was noted that a discharge of a total of 7 s duration was set to guarantee an open thermal equilibrium system so that the heating area could be reached prior to the initiation of measurement and all data collections were accomplished before the end of discharge.

The interference pattern was digitized in the frequency domain and post data processing involved three specific steps for both measurement and reference trace, which are displayed in Figure 2. Firstly, fast Fourier transform (FFT) was performed to convert both traces from frequency domain into time domain and obtain the scattering response as a function of time delay, which referred to Rayleigh pattern versus position. Secondly, applying an appropriate window in the transformed domain to extract the spectrum response of interest via inverse fast Fourier transform (IFFT). Finally, determining the wavelength shift by using ZNCC between two spectra separately from reference (room temperature) and measurement (glow heating).

The thermal sensitivity coefficient of vitreous silica fiber is estimated at 10.0 pm/°C for temperature calibration at telecom band [11,12]. As the reference trace was logged at room temperature around 22 °C, the estimated distributed high temperature profile as a function of fiber position are given in Figure 3, which varied with five applied currents and were shaped in Gaussian-like distributions. As to the straying dots in the results of 6.0 mA, they resulted from increasing errors of ZNCC derived from the thermal expansion and material transition occurring in the vitreous silica, which as a result deteriorated the correlation of the Rayleigh spectra between reference and measurement. It is noticeable that with the high density of released energy, the discharge created a great temperature gradient in the dimension of the sub-mm scale, which expanded as the applied current added up. The heat transferred to the fiber was primarily dissipated to the surroundings via radiation and thermal conduction down the length of the fiber. The maximum detectable value for the higher temperature measurement was up to 2100 °C and limited by the internal variation of Rayleigh spectra induced by the time-varying structure in the discharge duration, which was dominantly determined by the viscosity of the vitreous silica at different temperatures [8]. The result to some extent, coincided with the previous report on an optimum heating temperature of 2000 °C, near which, the surface tension and melted silica viscosity played roles in recovering fiber alignment so that less splice was obtained [4].

A moving window for retrieving RBS from the heating area contained 31 points, which corresponded to a spatial resolution of about 250 μm. The length of the heating area was up to 6 mm with a total of 10 mm converted by the moving window. Basically, the length of window used in IFFT for retrieving Rayleigh spectra of interest from reference and measurement determined the weight of contribution from local Rayleigh pattern. When a window of small length was applied, limited local contribution led to the insufficient spectral resolution to confirm the wavelength shift. Inversely, a larger length of window would average the detailed information derived from local spectrum. Generally, the straying dots in the result of 6.0 mA implied the little similarity of the two spectra in terms of correlation coefficient, which meant a great change occurred in between, for example, a softening and solidifying process.

Figure 4 indicates the relationship between the maximum temperature values of equivalent wavelength shifts and varied currents applied in multiple measurements, marked by black dots. A red dot represents a mean value for one current, with a standard error of mean for 20 individual tests (only 7 tests for 6.0 mA). Note that the plotted values were the ones with a correlation coefficient of wavelength shift above 50%, which ruled out the higher ones with less reliability than 50%. Detailed analysis of the correlation coefficient will be discussed later.

In summary, we demonstrated a new method for distributed high temperature measurement in a commercial fusion splicer based on optical frequency-domain reflectometry. The distributed temperature profile was attained by ZNCC between reference and measurement. The detectable peak temperature for optical fiber was equivalent up to 2100 °C at a nearly open thermal equilibrium system with approximate 6 mm length of thermal conduction area. It is, to our best knowledge, the first demonstration of the OFDR capability of ultrahigh temperature sensing with an upper limit of 2000 °C with the use of single mode fiber. Potential applications of this technique are measurement of rapidly changed temperature with sub-mm spatial resolution over 2000 °C for telecom fiber.

### 3.2. Statistical Analysis of Internal Stress Induced by Electrical Arc

Generally, a conventional static large strain test may not significantly change the Rayleigh scattering pattern as the discharge does. As the discharge process involved the heat transfer of high energy density from electrode to vitreous silica fiber, the measured reflected spectrum should have been affected consequently, which might give rise to optical loss and mechanical stress [13,14]. It probably came from the refractive index change arising from the optical absorption of localized electrons via ultraviolet radiation and thermal convection, energy transfer from electron motion to electron potential, thermal motion of lattice, thermal diffusion of dopants as well as phase transition from solid to liquid, and in between. Measuring the viscosity of the pure vitreous silica as a function of changing temperature might help to understand.

To investigate the internal stress arising from the great temperature gradient with natural cooling, multiple discharge tests were implemented at different currents. Local spectra of interest were extracted from mea#1, ref#1, mea#2, ref#2, mea#3, ref#3,…, mea#i, ref#i,…, mea#n, ref#n in order, in which the notation ‘mea#i’ represents the ith trace under discharge and ‘ref#i’ refers to the trace after the ith discharge. ZNCC was adopted in calculating the wavelength shift between spectra before and after discharge, such as ref#2⊗ref#1, ref#3⊗ref#2, ref#4⊗ref#3…, where symbol ⊗ refers to ZNCC operation on aforesaid spectra in the frequency domain, which was equivalent to internal formed stress. Note that none of the spectra was a fresh one without enduring the process of heating and cooling. The strain sensitivity coefficient was estimated to be 1 pm/με as a common value for @1550 μm telecom single mode fiber [11,15]. A statistical method was applied to evaluate the stress induced by the repeated heating and cooling process. We found that high temperatures above the softening point followed with natural cooling could result in an irreversible or permanent deformation in structure, which meant internal stress came into being as a product of nonuniform energy distribution and dissipation.

The standard deviation of wavelength shift as a function of currents ranging from 4.0 mA to 6.0 mA are shown in Figure 5a. It transpired that the fused area involved the formation of internal stress in which the wavelength shift arose gradually as current increased and showed a dramatic increase as the current added up to 5.5 mA. Figure 5b statistically exhibits the distribution of the correlation coefficients in a fused range that vary as a function of maximum wavelength shift under different currents, which indicated the similarity between the two Rayleigh spectra before and after discharge. In principle, the decrease in correlation coefficient is proportional to the displacement of two indistinguishable correlated Rayleigh spectra, which is plotted as an ideal case in dash-dot line. It is noted that as the current added up to 5.5 mA, this similarity began to collapse for the reason of being largely fused and dropped dramatically with increased wavelength shift, which conversely verified the deformation of fused silica due to the high temperature distribution. It was a real reflection of a large amount of stress modes being formed over a nonuniform temperature gradient within a small region over a short time. Hence the corresponding Rayleigh scatterings fluctuated at different positions with different wavelength dependence. Therefore, repeat discharge tests were implemented to illustrate the different situations of this nonuniform temperature distribution. As it was a random process, a Gaussian distribution was verified by this evolution process.

In summary, we proposed a correlation method to statistically analyze the stress that was induced near the softening point with respect to the wavelength shift of spectra. Stress under different levels of current were compared from repeated heating and cooling processes and it was noticed that negligible structure deformation occurred at 4.0 mA while accompanied with significant change above 5.5 mA. Using this method, we detected the irreversible deformation that confirmed the occurrence of phase transition in a fused range. Switching from the frequency domain into the time domain, other meaningful information that could be extracted in terms of delay shift, such as the irreversible expansion of the fiber length after repeated discharge, we discuss next.

### 3.3. Thermal Impact on Optical Path Length Indicated by Delay Shift

Ignore the dispersion effect of the ultrahigh temperature gradient in the silica fiber over the scanned range of OFDR, which is responsible for a very small change in the fiber section. The thermal-induced refractive index change and thermal expansion led to the changed optical path length over the heating range. The scattering response could be converted to the time delay compared with no arc condition. Theoretically, OPL reflects the delay of the scattering response in the time domain [16]. For OFDR in distributed temperature and strain measurement, accumulated delay shift is measured via correlation between changed and nonchanged measurement [12,15]. In practice, the spectral bandwidth of 100 nm imposed the delay shift accuracy at sub-picosecond. To improve it, the original signal was treated by zero padding prior to Fourier transformation into the time domain. Combining with ZNCC performing on the two Rayleigh patterns of interest between reference and measurement, the time resolving ability was enhanced by at least one order more than before. Detailed results are shown in Figure 6.

Delay shift is a scattering response of measurement compared to the reference in terms of delay time. We generally ascribe the delay shift to two contributions: the negative growth derived from the melt-induced reduction of the refractive index and the positive growth built up by the thermal-induced refractive index change and material expansion, which lead to the different scattering responses in between. Both factors are temperature dependent and provide different contributions to the overall delay. Note that there was a narrow delay interval centered around 18.11 ns, in which the delay shift dropped below the reference level and after it rose to a stationary level, as shown in Figure 6a. Compared with the previous result from the temperature profile in Figure 3, a deviation was noted between the time that maximum temperature and minimum delay shift approached. It could be the reason that the transition of localized electrons in silica occurred ahead of the thermal expansion of material as heat transferred from glow discharge to fused silica fiber. As a result, the weight center of negative growth moved ahead of the weight center of positive growth. Overall, the converted delay shift using the zero padding method fluctuated around ±20 fs, indicating the intrinsic fluctuation of system. Accordingly, the delay shift over the entire heating range for a single discharge was statistically calculated by the average value at both ends, which is shown in Figure 6b. It was noted that the relationship between delay shift for a single discharge and the applied currents was approximately linear.

For further investigation by using this temporal correlation method, as shown in Figure 7a, a multiple delay shift was initially observed from the measurement under relative weak glow heating, in which a new fiber was translated horizontally biased from the heating center with 4.5 mA current. The maximum value was near 1300 °C inferred from the converted temperature profile as shown in Figure 7b, which is below the softening point of fused silica. It was interesting to identify multiple scattering response shifts after light being through the heating center. Considering the chemical composition in Germanium-doped silica fiber at ultraviolet absorption band, the multiple delay shift could be derived from the Ge-dopant melted in the heating center while silica was still solid at this temperature range, which played a role of small-cavity leading to the multiple reflection of scattering light [17].

Basically, the selection of the reference for comparison determined what we observed from the variation of delay shifts. In the previous result, the delay shift was calculated in pairs, in which reference was not fixed but recorded subsequently after each measurement. As we expected, the thermal impact from the previous discharge on the fiber was reduced properly. Instead, fixing a reference at one current allows us to note the cumulative effect in terms of delay shift. As shown in Figure 8, vitreous silica fiber may remain close to an elastic solid or maintain low expansion properties below the melting temperatures corresponding to discharge current around 4.0 mA, 4.5 mA, since the corresponding delay shift fluctuated within the computational error after enduring repeated heating and cooling. Conversely, in the cases of 5.5 mA and 6.0 mA, the delay shift was accumulated as the number of the discharges increased, which was exactly implied by the standard deviation of wavelength shift shown in Figure 5a. It was also plausible if referring to the viscosity of vitreous silica above softening point at 1670 °C whereby the weight of the fiber itself and external impact from discharge probably played roles in deformation or elongation of the fiber [8].

In summary, based on temporal correlation between scattering response in reference and measurement, distinct variation of the delay shift induced by the phase transition of fused silica was observed in the duration of discharge. It also showed an advantage over the spectral correlation, such as the detection of multiple delay shifts due to the coexisting phase modes that could not be conferred from wavelength shift. Lastly, a cumulative delay shift was extracted from the repeated discharge tests by using this method, which indicated the extent of OPL that may raise errors in the spectral correlation method.

### 3.4. Analysis of the Correlation Coefficient

The correlation coefficient indicated the similarity between two spectra in comparison, and the reliability of the value of the wavelength shift or delay shift we obtained by the ZNCC method. By comparing the coefficient, we found an assessment to quantitatively evaluate how reliable the values were for converted temperature or delay shift. Therefore, a coefficient above 0.9 represented strong similarity of two spectra or patterns while below 0.5 could mean high deformation of one of the spectra or patterns due to the transition of fused silica. The percentages of the coefficient above 50 or 60% were calculated accordingly, the value of which generally gave the reliability of the temperature and delay shift measurement at different currents, as shown in Figure 9c,d for temperature measurement and Figure 10b for delay shift measurement.

In former temperature measurements, the wavelength shift between the two Rayleigh spectra were calculated by ZNCC and converted into a temperature value with an assumed thermal coefficient, which is shown in Figure 3. Since we had already obtained the wavelength shift and corresponding correlation coefficient from multiple tests, statistical analysis was followed by grouping the coefficients with respect to different currents as a function of wavelength shift. Intuitively, since high energy density generated by electrical discharge gives rise to absorption in the ultraviolet band of vitreous silica, the light propagation and scattering response should be influenced directly by the local refractive index change and thermal expansion. For contrast, the averaged correlation coefficient versus wavelength shift was separated into two groups corresponding to different spatial areas of the heated fiber region: left-hand side, shown in Figure 9a and right-hand side, shown in Figure 9b. On the one hand, it denoted that the similarity between two spectra descended concussively with ascending temperature, which was characterized by wavelength shift. On the other hand, inferred from the size of the error bar between the two sides, the variability of the coefficient at the right-hand side behaved more divergently than that of the left-hand side. It presumably indicated the increase of the optical path length variability of the fiber section through which the light was propagating, induced by the electric discharge. Similar data processing was done for statistical analysis of delay shift measurement, which is shown in Figure 10a.

In summary, we gave an assessment of the coefficient obtained by the ZNCC method and quantitatively evaluated the reliability of wavelength shift and delay shift based on the percentage of the coefficient above 50 or 60% at different currents.

### 3.5. Dynamic Impact on OPL and Local Rayleigh Spectrum

Optical fiber fusion splicing involves a series of basic tasks and processes, which have been discussed in [18]. Here, we proposed a spectral mapping method for monitoring the dynamic impact on the OPL as well as local Rayleigh scattering responses. By sacrificing the temporal resolution that the length of moving window determined, we could achieve spectrum monitoring in terms of scattering response within a single scan. Compared with the real-time monitoring of OTDR, this approach was not limited by the repetition rate of pulses since Rayleigh spectra retrieved by IFFT exclusively assembled the amplitude and phase components that different weights of the local Rayleigh pattern held within the selected window at each moving step. Continuous variation of intensity in terms of local reflection or scattering response were monitored by comparing to the reference one.

The dynamic impact of measurement on OPL was assisted by monitoring the fiber end in terms of its transmission and delay time that accumulated from the discharge area. Regardless of the wavelength dependence of the refractive index over the wide range of temperature, we could generally ascribe this dynamic impact to the radiation-induced dynamic change of the refractive index, heat transfer and thermal expansion of silica fiber. High reflection of the fiber end helped improve the contrast of time-varying information, which inversely denoted the time-varying OPL that mostly derived from the discharge impact. The processes which involved different durations of discharge are shown in Figure 11a−d. There was a large fluctuation in Rayleigh scattering signal around melting temperature, in which ∂ε(z)/∂t≠0 due to a large change in the refractive index among softening and solid regions of fused silica over a short time. As a result, the noncontinuous changed transmission was measured as expected, as it represented a rapid electronic response to the radiation of the arc that happened at much faster speed than the response time of OFDR. For the following gradual variation of transmission relaxing to a stationary level of delay time, it might refer to the relaxation time of phase transition that was taking place during the heat transfer or diffusion process, which led to the expansion and contraction of the material.

Splicing mode involved prefusion cleaning, fusion splicing, refusion. In our test, push action was set to be silenced and every single scan was accomplished in the duration that critical moment happened. As shown in Figure 11a, two actions were involved by the order in which predischarge and normal discharge were implemented. For decomposing and vaporizing any debris attached to the fiber, it first created a relatively high temperature environment that lasting for 0.2 s within a small space, then followed by an extremely high temperature that normal discharge built up in order to rise to the softening point.

For the duration of predischarge, the OPL decreased as the electric arc released, in which radiation absorption led to the fast transition of electrons from ground state to excited state and hence the drop of refractive index. It could be possible that delay time reached 27.275 ns as a minimum before the thermal motion of the lattice began to dominate, which led to the expansion of the physical length gradually, and extended the initial delay time from 27.305 ns to 27.310 ns. Followed by the predischarge with 0.5 s gap, the arc of normal discharge created an extremely high temperature such that the refractive index dropped instantly due to the rapid response of the electrons by radiation absorption, and the delay reached the minimum. Inferred from the relaxation time of delay from minimum to equilibrium, a tenth of a second is presumably required to transfer heat from the outside to the inside of the fiber and establish a uniform temperature field. For the case of the redischarge process shown in Figure 11b, the thermodynamics process was similar to the previous action modes. The only difference was that without predischarge, it raised two discontinuities in terms of delay time, which might involve an obverse and reverse phase transition between local solid and local liquid. In the duration of discharge, shown in Figure 11c, a dynamic equilibrium state was maintained in an open thermal system. Compared with the result obtained at room temperature, it was almost 0.2 ps longer, which confirmed the delay shift discussed before. After termination of discharge, a discontinuity shown in Figure 11d represented the phase transition from local liquid to solid accompanied with a relaxation time.

Corresponding to the variation of OPL in the duration of redischarge in Figure 12a, the monitoring of dynamic impact on the Rayleigh scattering response is shown in Figure 12b. It was noted that inside the red dot lines a distinct rise of intensity was observed from the local Rayleigh spectrum by simply subtracting the monitoring result of measurement from reference. Presumably, it could be the generation of the solid−liquid boundary that a high reflection condition was exactly formed as instant release of arc kicking on the heating center, which severely brought down the refractive index locally and led to the spark of Rayleigh scattering. The delay of spark also responded forwardly due to the reduction of OPL, while the area inside the black dotted lines was the trace that multiple discharges left before. Generally, the spark could indicate the softening of fused silica, which helped testify to the limited current used for perfect splicing.

In summary, we proposed a spectral mapping method for investigating the dynamic impact of arc on the fiber with respect to the OPL and local scattering responses at different stages of discharge, which showed a spectral monitoring ability. It was noted that, based on the monitoring results obtained by this method, the delay induced by the change of refractive index had tradeoffs with that induced by thermal expansion under an open thermal equilibrium of discharge, and to some extent, delayed response of the fiber end as a function of time indicated the relaxation time of phase transition, which lasted tenths of seconds. Analysis on the discontinuity of phase transition in the release and extent of the arc were also given. In addition, dynamic impact on the Rayleigh scattering pattern was found at the beginning of the redischarge, this phenomenon disclosed the melting or softening of vitreous silica at a certain moment.

## 4. Conclusions

Generally, we provided a feasible method for distributing ultrahigh temperature measurements in a dimension of sub-mm scale based on OFDR, and investigated the thermal impacts starting from initiation, duration, and termination of discharge by using a fused single mode fiber. The distributed temperature profile was obtained by an assumed thermal coefficient of 10 pm/°C @1550nm, with maximum detectable temperature for fused silica up to 2100 °C under a nearly open thermal equilibrium state. The effects on the statistical analysis of the internal stress induced by electrical arc at different currents presumably gave a general estimation of the mechanical strength induced by repeated discharge. It is interesting to note that by correlating the Rayleigh patterns in time domain, the delay shift under discharge was observed accordingly at different currents with ±20 fs of system fluctuation, which may provide a promising method on the group birefringence measurement, an exploration of phase transition in materials. Evaluation of the reliability of the measurements has been explored by statistically analyzing the correlation coefficient, indicating the limitation of OFDR on the high temperature measurement. In the end, by adjusting the temporal resolution of Rayleigh scattering response in time domain, a spectral mapping method is proposed for spectral monitoring of the dynamic impact on OPL and local Rayleigh scattering response, which displayed a good advantage over the real time monitoring of OTDR by not being limited by the repetition rate of pulse.

## Figures and Tables

**Figure 1 sensors-20-06407-f001:**
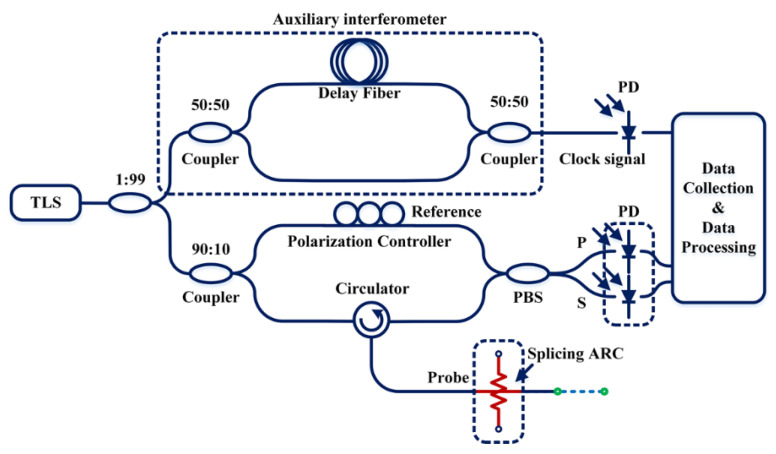
Schematic of the optical frequency-domain reflectometry (OFDR) system for electrical arc discharge measurement applied with different currents.

**Figure 2 sensors-20-06407-f002:**
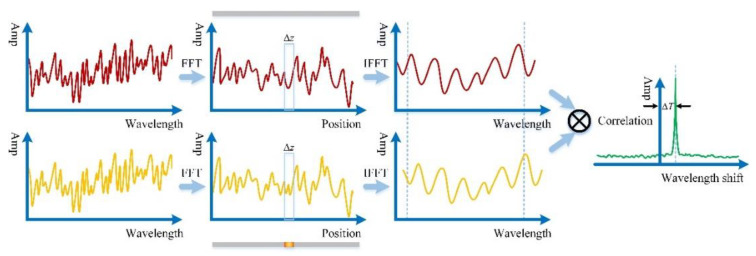
Demonstration of general method of calculating a thermal shift.

**Figure 3 sensors-20-06407-f003:**
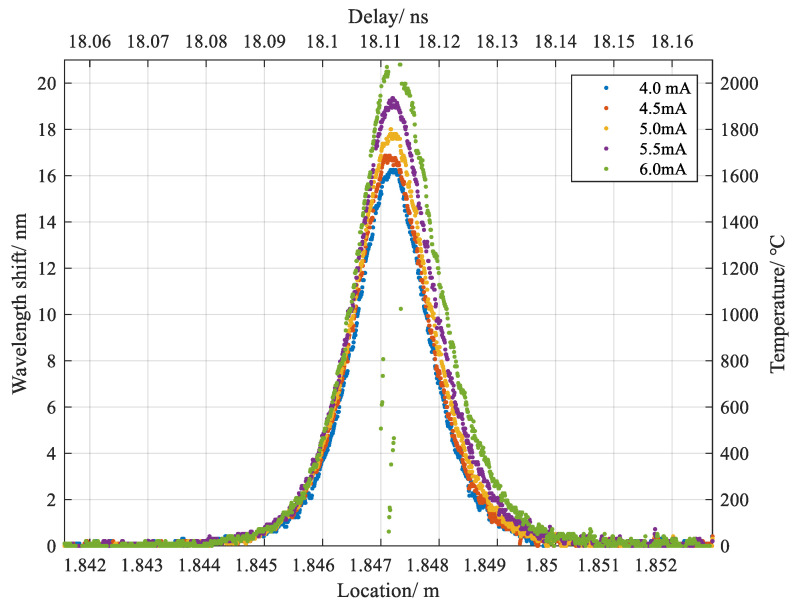
Distributed temperature profile varied with five selected currents along the fiber.

**Figure 4 sensors-20-06407-f004:**
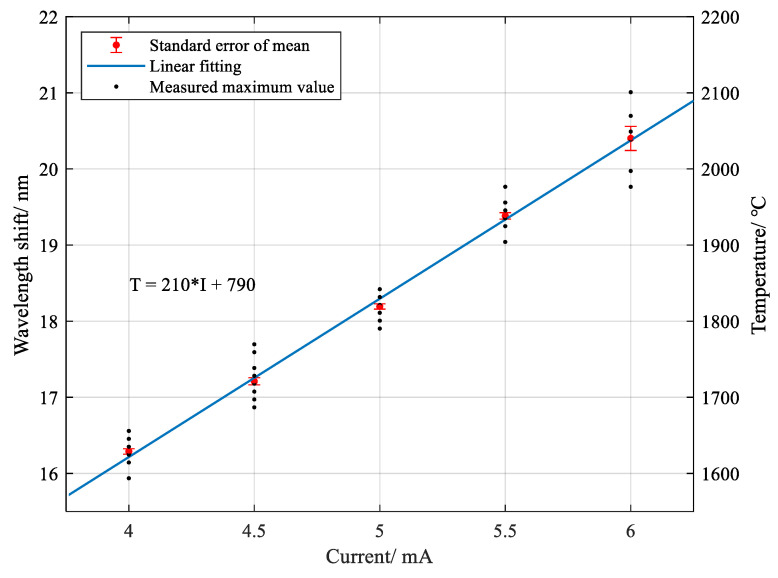
Relationship between wavelength shift (temperature) and applied discharge currents.

**Figure 5 sensors-20-06407-f005:**
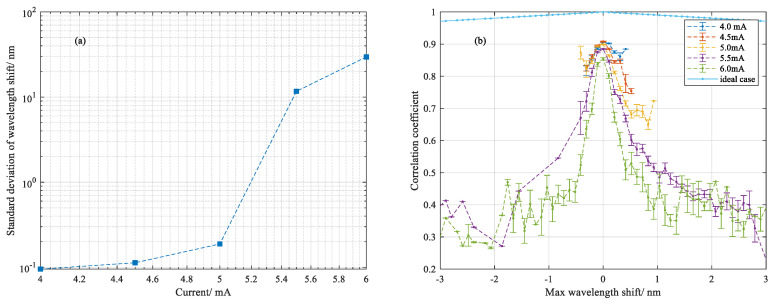
(**a**) Standard deviation of wavelength shift in fused range as stress varied with applied arc current; (**b**) distribution of correlation coefficients for maximum wavelength shift at different currents.

**Figure 6 sensors-20-06407-f006:**
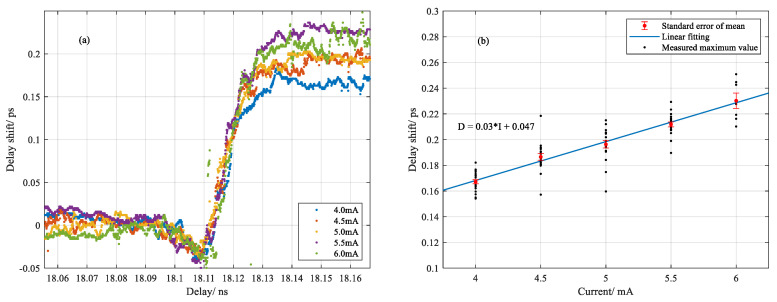
(**a**) Variation of delay shift with different current; (**b**) relationship between delay shift (equivalent change of OPL) and applied discharge currents.

**Figure 7 sensors-20-06407-f007:**
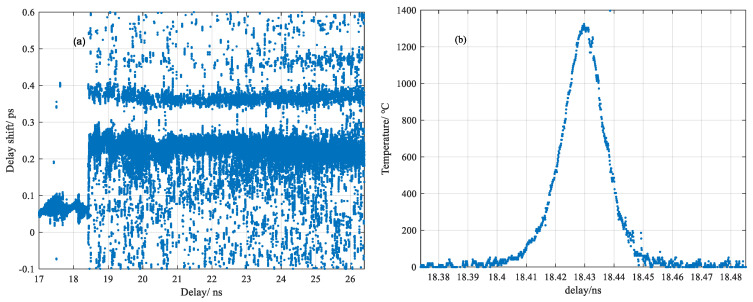
(**a**) Multiple delay shifts corresponding to the (**b**) converted temperature distribution biased from the heating center.

**Figure 8 sensors-20-06407-f008:**
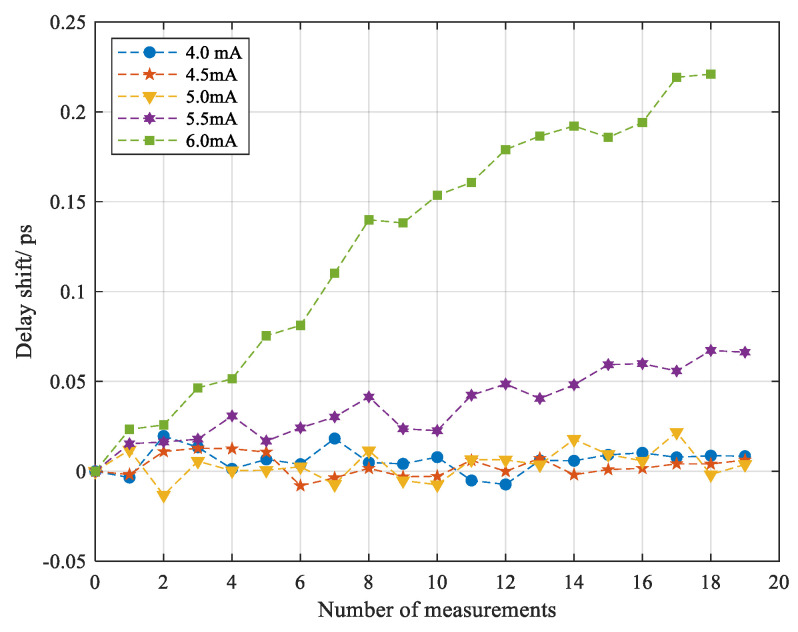
Cumulative delay shift varied with the number of discharges for different currents.

**Figure 9 sensors-20-06407-f009:**
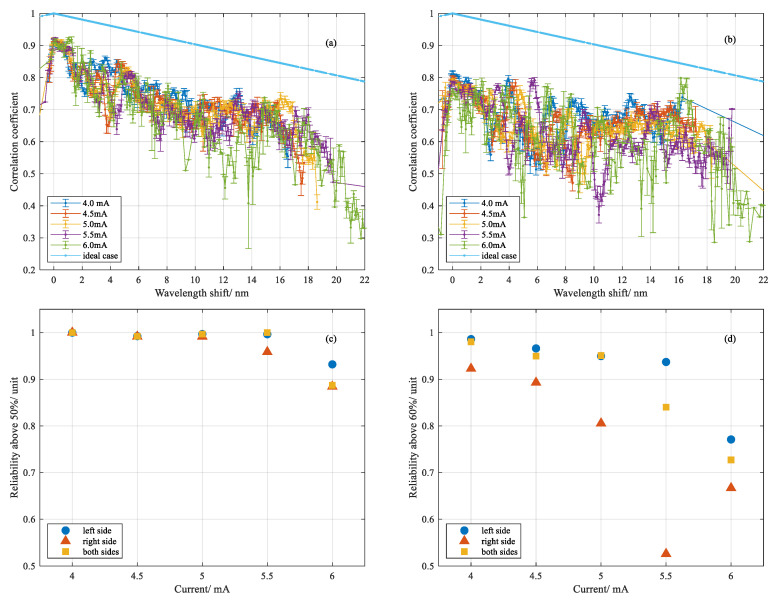
Statistical analysis of the coefficient versus wavelength shift and percentage of coefficient versus current: (**a**) coefficient versus wavelength shift at left-hand side; (**b**) coefficient versus wavelength shift at right-hand side; (**c**) percentage of coefficient of wavelength shift above 50% for different currents; (**d**) percentage of coefficient of wavelength shift above 60% for different currents.

**Figure 10 sensors-20-06407-f010:**
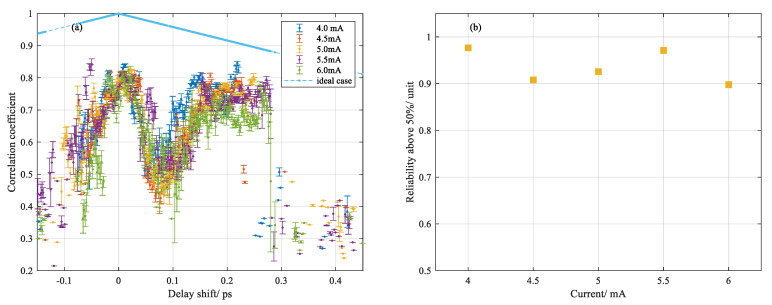
(**a**) Coefficient varied with maximum delay shift and (**b**) percentage of coefficients above 50%.

**Figure 11 sensors-20-06407-f011:**
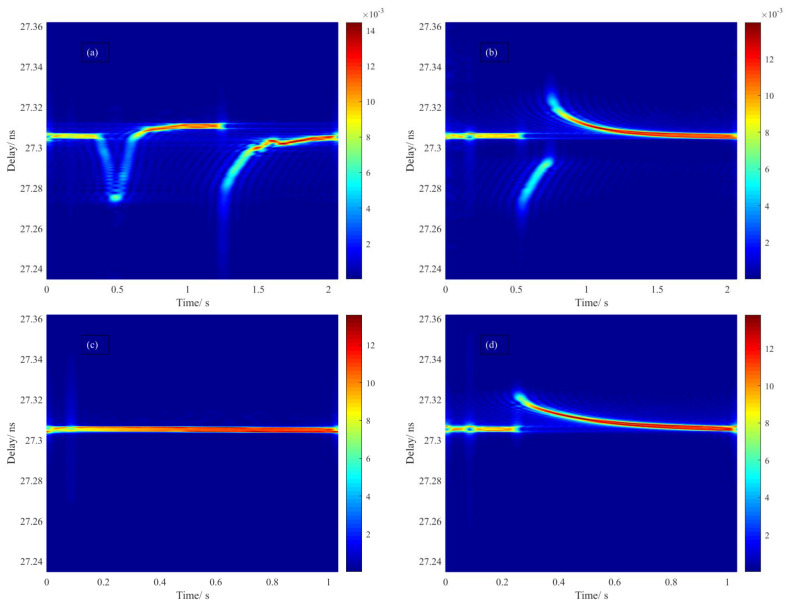
Transmission spectrum at different stages of discharge: (**a**) initiation of predischarge followed by normal discharge; (**b**) initiation of redischarge; (**c**) duration of discharge; (**d**) termination of discharge.

**Figure 12 sensors-20-06407-f012:**
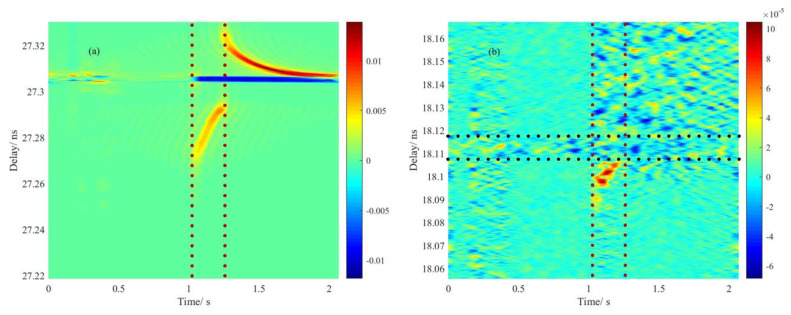
Demonstration of dynamic impact on local Rayleigh spectra indicated by the transmission spectra of the fiber end: (**a**) transmission spectra of the fiber end in the duration of redischarge; (**b**) Rayleigh spectra located at redischarge center.

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
