# Peer review of "Distributed High Temperature Monitoring of SMF under Electrical Arc Discharges Based on OFDR"

_sensors, 2020, doi:10.3390/s20226407_

Round 1
Reviewer 1 Report
This work offered an intensive distributed temperature monitoring of SMF under arc discharge by utilizing the fusion splicer differently. It is recommended for publication after considering the following comments:
Comment 1: Line 15-17. The sentence “Estimation … coefficient” is difficult to understand
Comment 2: Line 39-43. Too long to be understood
Comment 3: Line 115. Please explain a little bit about “artificially controlling the commence of data collection”
Comment 4: Line 171, Figure 4. Are these data points at each current obtained by continuous measurement as you did in Figure 6, OR the data come from separate series of experiments (from 4 mA to 6 mA).
Comment 5: Line 257. Since it is approximately linear, giving a linear fit for Figure 6(b), also for Figure 4 if possible.
Comment 6: Give citations to the assumptions for: thermal coefficient of 10 pm/°C; strain sensitivity coefficient 1 pm/µε for @1550 nm telecom SMF. (there is a typo in Line 198, change µm to nm)
Comment 7: related to commend 4, if the data is obtained continuously rather than from different experiments. Can the results be universally obtained? I mean, if the stripped SMF used in this investigation be replaced by another SMF, could you obtain similar results? One concern is, the arc is used to fuse fibers so it may irreversibly deform the fiber, making the observation not repeatable.
Comment 8: some minor issues: Please type the symbol “°C” correctly throughout the manuscript. The character code is “00B0” in “Symbol”; Line 248: “Figure 6(a)”, but Line 258 “Figure 7 (b)”, please be consistent in the paper.
Author Response
Dear reviewer,
We did our revision of the paper, for the detailed content, please see the attached file.
Best regards,
Chen.

Reviewer 2 Report
The authors report the application of an Optical frequency-domain reflectometry (OFDR) technique to obtain the temperature distribution of a SMF under electrical arc discharges. The topic is quite interesting since fusion splicers are commonly used in the development of fiber-based sensors and the knowledge of the process is relevant to its success and obtain good repeatability. In general, the paper is well structured, and all the sections complement each other. Figures are also clear and adequately complement the information mentioned in the text. Methods are well described and the results, and corresponding analysis, support conclusions. The major drawback is the English writing. A full revision is necessary. Some examples are mentioned bellow, besides some other points to improve.
- The phrase in line 28 must be corrected. For example, use instead: “Several works have been reported regarding high splicing temperature measurements in electric arc discharges”.
- In line 51, use “determines the softening and annealing temperatures of the material” instead of “determines the temperature of softening and annealing of the material”.
- In line 74, correct “of fiber under test” by “of the fiber under test”. Missing “the” and “a” is common along the text and I will only mention some.
- In line 87, “Knowing that the dispersion” instead of “Knowing the dispersion”.
- In line 97. The phrase becomes clear if altered as:
“(…) makes the former approximation for the refractive index no longer valid (…)”.
- In line 112, the phrase seems “displaced”. Perhaps the meaning becomes clear having:
“As the spectrum shift is proportional to the stimulus applied on the fiber, these (…)”
- In line 119, the phrase should be: “In our setup, the tuning rate of the light source was set to (…)”
- Figures should appear after being mentioned in the text. The authors should make sure this happens.
- I found the phrase in lines 143-145, explaining the straying dots in figure 3, somehow confusing. A possible improvement is:
“As to the straying dots in result of 6.0mA current, it resulted from increasing errors of ZNCC derived from the thermal expansion and material transition occurred in vitreous silica, which as result, deteriorated the correlation of Rayleigh spectra between reference and measurement.”
- Line 156: “(…) a spatial resolution of about 250μm. The length of heating area (…)” instead of “(…) a spatial resolution about 250μm. The length of heating area (…)”
- Line 157 (and elsewhere it appears), “Basically, the length of window” instead of “Basically, length of window”.
- Line 215, “(…) that is induced (…)” instead of “(…) that induced (…)”
- Line 222, “we will discuss it in the next” should be corrected by “we will discuss it next”
- Line 239, “against with” is incorrect. Use just “with”.
- Line 361, replace “In duration of pre-discharge” by “For the duration of the pre-discharge”
- Line 418, use “by not being limited” instead of “that not limited”.
Author Response

(The authors gave the same response as above.)

Reviewer 3 Report
The authors present and experimentally demonstrate a distributed high temperature measurement by measuring the thermal-induced wavelength shift based on OFDR. Result and data analysis indicates some new findings, which I think could be useful to scientific community. However, several issues are required to be addressed before acceptance.
- As a system that relies heavily on data processing and analysis, the manuscript has only one equation. Internal stress, thermal-induced wavelength shift, delay shift, et al. need more explanation and show their relevance. Could the author spend more effort on the principle and provide readers a thorough explanation of the proposed findings?
- Authors claim that correlating Rayleigh patterns in the time domain may provide a promising method on the local birefringence measurement and exploration of phase transition in material. But there is no more information can verify this content in the manuscript. I suppose authors should make further discussion.
- As far as I understand in this manuscript, the experimental conditions of the distributed high temperature monitoring system are rather complex and high cost, which means great challenges to achieve long distance sensing and practical implementation. Therefore, I have a suspect that the proposed spectral mapping method allows a significant superiority over the mature real time monitoring of OTDR technology.
- In addition, there are obvious language and format errors in the manuscript. The serial number of the subtitle does not match that of the main title in Section 3. Moreover, 5 parts of data analysis should be integrated to main points, to clarify the significance of each in this research.
Author Response

(The authors gave the same response as above.)
